# The Coronavirus Footprint on Dual-Task Performance in Post-Acute Patients after Severe COVID-19: A Future Challenge for Rehabilitation

**DOI:** 10.3390/ijerph191710644

**Published:** 2022-08-26

**Authors:** Marica Giardini, Ilaria Arcolin, Marco Godi, Simone Guglielmetti, Alessandro Maretti, Armando Capelli, Stefano Corna

**Affiliations:** 1Istituti Clinici Scientifici Maugeri IRCCS, Division of Physical Medicine and Rehabilitation of Veruno Institute, 28013 Gattico, Italy; 2Istituti Clinici Scientifici Maugeri IRCCS, Division of Pulmonary Rehabilitation of Veruno Institute, 28013 Gattico, Italy

**Keywords:** COVID-19, dual-task, balance assessment, TUG test, healthy subjects, rehabilitation, falls risk, prevention

## Abstract

Recent studies suggest that also the non-critical form of COVID-19 infection may be associated with executive function impairments. However, it is not clear if they result from cognitive impairments or by COVID-19 infection per se. We aimed to investigate if patients in the post-acute stage of severe COVID-19 (PwCOVID), without manifest cognitive deficits, reveal impairments in performing dual-task (DT) activities compared to healthy controls (HS). We assessed balance in 31 PwCOVID vs. 30 age-matched HS by stabilometry and the Timed Up and Go (TUG) test with/without a cognitive DT. The DT cost (DTC), TUG test time and sway oscillations were recorded; correct cognitive responses (CCR) were calculated to evaluate cognitive performance. Results show a significant difference in overall DT performance between PwCOVID and HS in both stabilometry (*p* < 0.01) and the TUG test (*p* < 0.0005), although with similar DTCs. The main difference in the DTs between groups emerged in the CCR (effect size > 0.8). Substantially, PwCOVID gave priority to the motor task, leaving out the cognitive one, while HS performed both tasks simultaneously. Our findings suggest that PwCOVID, even without a manifest cognitive impairment, may present a deficit in executive function during DTs. These results encourage the use of DTs and CCR in PwCOVID.

## 1. Introduction

COVID-19, caused by the severe acute respiratory syndrome coronavirus 2 (SARS-CoV-2), has a scope and severity unparalleled in modern society. There is increasing evidence that coronaviruses spread to extra-respiratory organs, notably the central nervous system: human coronaviruses are one of several viral groups that are considered able to penetrate the brain and cerebrospinal fluid [1]. Neurological manifestations in people with COVID-19 (PwCOVID) were demonstrated from the beginning of the pandemic [2]; a percentage between 5% and 40% of PwCOVID presented some neurological manifestations, with central involvements being more common [3]. In particular, PwCOVID may present disorders of consciousness, acute cerebrovascular disease and, predominantly, cognitive impairment [4,5,6].

Based on neuroimaging data, the temporal and frontal lobes appear to be particularly vulnerable to COVID-19 and are involved more frequently than any other structures [7,8], resulting in lasting memory impairment and executive deficit, respectively. The brainstem has also been implicated, potentially resulting in physiological arousal impairment [9].

Neuropsychological studies seem to confirm the findings of neuroimaging. In fact, a recent review found that subjects, following severe to critical COVID-19 infection, suffer from short-term cognitive impairment compared to healthy controls [10]. Although a large proportion of subjects included in this review had a global cognitive impairment, a cognitive profile of subjects characterized by a normal score is, however, recognizable in common global cognitive screening tools (such as the Montreal Cognitive Assessment (MoCA) or the Mini Mental State Evaluation (MMSE)), with a tendency for lower performances in executive than in other cognitive function [11,12].

Cognitive problems may not be limited only to PwCOVID who are critically ill [13]. Recent studies suggest that also the non-critical form of COVID-19 infection may be associated with cognitive impairments, especially in the domain of executive functioning, which can persist for some months, even in young individuals who did not require hospitalization [14,15]. While deficits in executive functions may not represent a serious clinical problem in young people, they could have implications in older people, since they are associated with an increased risk of falls [16]. Although there is no consensus on what the most appropriate test is to use for fall risk assessment [17], examining people during a gait or a balance task while they perform a secondary task (dual-task paradigm) is an accepted way to assess the interaction between cognition and mobility [18] and, consequently, to assess the risk of falls [19]. It has been demonstrated that a slower walking speed while counting backwards is associated with recurrent falls in frail adults [20]. Moreover, assessing dual-task performance is also clinically important, since in daily activities the ability to perform two tasks simultaneously is an essential skill for independent living [21] and may better describe the potential for adverse events such as falling [22]. When two single tasks (cognition plus motor) are simultaneously performed, more cognitive resources are required, reflected by a higher brain activation in the prefrontal cortex, compared with single tasking [23,24]. Since, among motor skills, balance is acknowledged as a major predictor of falls [25], we aimed to evaluate both the dynamic and static components of balance in dual-task vs. single-task conditions.

To assess dynamic balance, the Timed Up and Go (TUG) test is commonly used [26], both with and without the dual-task component [27]. To assess static balance, stabilometric platforms can provide objective information useful for improving the quality of the healthcare treatments provided [28]. In fact, the center of pressure recorded by posturography is frequently used to characterize postural sway and studies have shown that it correlates with poor balance and risk of falls [29]. It can reveal subclinical balance disorders undetectable by a routine clinical assessment, even in people minimally impaired or unaware of balance impairment, and it is enhanced by the dual-task [30].

To date, the dual-task performance of people with COVID-19 in the post-acute stage of disease has been examined only in one study [31]. Specifically, Morelli et al. [31] investigated if patients recovering from severe and critical COVID-19 presented dual-task mobility deficits comparable to those of individuals with chronic lung disease. The authors found that the mobility performance during a cognitive dual-task was linked to illness severity; in fact, patients with an history of critical illness showed worse performance with respect to those who had only a severe form of COVID-19. Nevertheless, most of the patients enrolled in that study were affected by mild cognitive impairment, with a higher percentage of affected people in the critical illness group. Consequently, the results of the mobility performance with the addition of a cognitive task may be mostly related to the cognitive impairment rather than COVID-19.

Thus, the aim of the present study was to evaluate the effects on balance performance in PwCOVID by adding a second demanding cognitive task such as counting backwards, comparing the results with healthy subjects (HS). Our hypothesis was that PwCOVID in a post-acute phase after severe COVID-19 would show difficulties in the dual-task, with a significant reduction in performance compared to HS, even if their results were normal on cognitive tests. If this hypothesis was confirmed, these impairments, not visible on first examination, would have important implications for the discharge home of these patients and the rehabilitation required to recover a premorbid status.

## 2. Study Design and Methods

All patients admitted to the COVID-19 Rehabilitation Unit of Istituti Clinici Scientifici Maugeri IRCCS (Gattico-Veruno, Piedmont, Italy) between March 2021 and May 2021 with a diagnosis of severe COVID-19 in the acute phase [32] were screened for inclusion. They still had a positive swab and were assessed in a temporary lab on day 25 of admission to the ward, as the maximum allowed length of stay in Piedmont (Italy) in the COVID-19 rehabilitation ward was 30 days. Pre-existing data of HS, age-matched to PwCOVID, were extracted from the database of the Laboratory of Posture and Movement. The HS had been recruited in previous years for various research reasons not strictly related to this study, but they underwent the same evaluations, carried out by the same assessors and with the same instruments, as the current PwCOVID. Therefore, while the sample of COVID-19 was a convenient one, the group of HS was a stratified random sample generated from our database.

The inclusion criteria for performing the tests were similar for both HS and PwCOVID: (1) age >18 years, (2) total score on the MMSE >24 points (normal cognitive performance [33]), (3) ability to walk independently, (4) written consent to participate in the study. Hence, people with orthopedic problems that affect gait, neurological or respiratory disease, in non-stable clinical conditions, or whose hospital stay had been exclusively in the intensive care unit were excluded from the present study. People taking drugs known to affect sensory-motor function, affected by diabetes mellitus or with medium-to-severe sensory alteration of the lower limbs were also excluded. Patients were neuropathologically assessed with the neuropathy impairment score (NIS), which includes the ankle reflex, vibration, pinprick and temperature (cold tuning fork) sensations at the big toes. Bilateral lower limbs were independently evaluated (maximum score: 10 points). People showing diabetic peripheral neuropathy (NIS score ≥6 [34]) were excluded. For all participants, general characteristics such as age and sex were collected. Weight and height were measured and used for the calculation of the body mass index (BMI) with the following formula: “BMI = weight/height^2^”. Finally, the presence of comorbidities was assessed through the Cumulative Illness Rating Scale (CIRS) [35].

All participants signed a general consent form allowing future use of their records for medical research and all evaluations were approved by the institutional Ethics Committee (approval number 2592 CE).

### 2.1. Test Procedure

Before the evaluations, each test was explained to the participants. To minimize methodological bias, all measurements were made by the same researchers and all participants were given the same instructions. In the dual-task condition, participants were instructed to perform the two (cognitive/motor) tasks simultaneously but without giving specific priority to either of the two tasks. To avoid other potential biases, the single-task (ST) and dual-task (DT) measurements were performed in random order.

PwCOVID performed the tests carrying the device for delivering oxygen if oxygen supplementation was prescribed at the time of assessment. In each test, the Modified Borg Scale for Perceived Dyspnea at rest was administered [36] in order to check patients’ fatigue, and transcutaneous arterial O_2_ saturation (SpO_2_) was continuously monitored by pulse oximetry. Patients’ breathing rate was measured in order to calibrate and check the test stress [37]. A 5 min resting period was included between the stabilometry and TUG tests.

### 2.2. Motor Task

(a) Stabilometry was conducted with patients standing on a force platform (Medicapteurs, France), which recorded the sway of the center of foot pressure (CoP) during quiet stance. Patients stood barefoot with eyes open (EO) or eyes closed (EC). Feet were placed at an angle of ± 15° from the sagittal plane with a distance between heels of 2 cm. For each visual condition, two trials, each lasting 51 s [38], were performed. Sway area (SA) is the 95% confidence ellipse of the dispersion of CoP data, and sway path (SP) is the distance covered by the moving CoP. The data of the two trials (for both EO and EC) performed in the same condition were averaged; SA and SP were recorded as an indicator of the patient’s motor performance.

(b) The TUG test is the shortest, simplest clinical balance test available to predict risk of falls [39,40]. Participants performed an initial practice trial to familiarize themselves with the procedure, and then executed the TUG test twice. The time to perform each trial was recorded with a stopwatch and the values were averaged; time was recorded as an indicator of the patient’s motor performance. All subjects performed the test without walking aids.

### 2.3. Cognitive Task

For the single cognitive task, patients were seated comfortably in a quiet room. They were asked to count backwards in threes for 30 s [41], starting from a random number between 200 and 230, whilst gazing at an eye-level target 50 cm from the eyes. This serial subtraction task measures attention, mental calculation and working memory [42]. Participants were instructed to continue counting, even if they were aware they had made a mistake.

When the time was up, the total number of answers and the number of correct answers were recorded. As an indicator of cognitive performance, the correct cognitive response (CCR) [43] was calculated (see formula below (1)), where the number of ‘total subtractions’ includes both correct subtractions and mistakes. In the formula, higher scores are associated to better cognitive performances:(1)$$\mathrm{CCR}=\frac{\#\;\mathrm{Correct}\;\mathrm{subtractions}\;}{\mathrm{Trial}\;\mathrm{duration}\;\left(\mathrm{seconds}\right)}\times \frac{\#\;\mathrm{Correct}\;\mathrm{subtractions}\;}{\#\;\mathrm{Total}\;\mathrm{subtractions}}$$

### 2.4. Dual-Task

Subjects performed both the stabilometry assessment and the TUG test whilst simultaneously performing the calculation task. Subjects were instructed to focus on both the motor and the cognitive task at the same time, continuing to count even if they were aware of mistakes made. For each trial, the CCR and the dual-task cost (DTC) were calculated. As there is at present no gold standard for DTC calculation, we used a common formula:(2)$$\mathrm{DTC}=\frac{\mathrm{Dual}\;\mathrm{task}\;-\;\mathrm{Single}\;\mathrm{task}}{\mathrm{Single}\;\mathrm{task}}\times 100$$

Expressing the result as a percentage, where higher scores correspond to higher cost due to the addition of the second task. DTC is defined as the decline in DT performance compared to ST performance of a task [44].

Summarizing, TUG time, sway path, sway area and the DTC were recorded as indicators of motor performance during DT execution; on the other hand, the CCR was used as an indicator of cognitive performance during the DT.

### 2.5. Sample Size

Until now, there has been no study that focused on the comparison of dual-task performance of PwCOVID with that of HS. Therefore, in order to calculate our sample size, we relied on the single study that investigated the dual-task performance of PwCOVID, even if it compared PwCOVID with those with chronic lung disease [31]. Nevertheless, we are aware that this choice has led to a small overestimation of the sample, since the expected difference between HS and PwCOVID should be greater than that between chronic lung disease and PWCOVID [45].

Specifically, Morelli et al. [31], in their assessment of DT performance during the TUG test in PwCOVID, found a difference of 3.8 ± 5.2 s between individuals with COVID-19 vs. those with chronic lung disease. This corresponds to a Cohen’s *d* effect size of about 0.7. Setting the significance at 5% and statistical power at 80%, we therefore required a sample size of 27 subjects for each group [46].

### 2.6. Statistical Analysis

Mean ± standard deviation (SD) values were used for descriptive statistics, and mean ± standard error (SE) for the figures. Differences between groups (PwCOVID vs. HS) in clinical variables, assessment scales, DTC and CCR were detected by the Student’s *t*-test and Chi-square test, as appropriate.

For stabilometric variables (SA and SP), we performed ANOVA with the two groups as independent variables and within repeated measures (respectively, EO and EC; and ST and DT). For the TUG test, a two-way repeated-measures ANOVA was run between the two groups (PwCOVID and HS) and within the two task conditions (ST and DT). In all analyses, the Bonferroni correction was applied to compensate for alpha inflation due to multiple comparisons, and for statistical significance, a value of *p* < 0.01 was set. When the result was significant, Tukey’s post hoc test was run.

In order to investigate the clinical meaning of differences between PwCOVID and HS, Cohen’s *d* effect size was calculated, with a small effect defined as 0.2, a medium effect as 0.5 and a large effect as ≥ 0.8 [46]. The linear relationships between CCR EO and CCR EC in PwCOVID and HS in stabilometric assessment were computed and a *t*-test was calculated on the regression slopes. Statistical analysis was performed using Statistica software (StatSoft Inc., Tulsa, OK, USA).

## 3. Results

Of 117 PwCOVID screened, 31 (27%) met the inclusion criteria and were recruited. PwCOVID had a length of stay in acute care of 11.9 ± 9.4 days; nine had unilateral lung pneumonia, and 22 had bilateral lung involvement. Continuous Positive Airway Pressure (CPAP) was required in 10 patients, while four received oxygen via a Venturi Mask and 17 simply via a nasal cannula. However, at the assessment time, only six PwCOVID (24%) had a prescription of oxygen supplementation. Table 1 presents the clinical characteristics of PwCOVID and HS: no significant difference was found between groups in terms of sex, age, weight, height, BMI, NIS and MMSE. Before the study protocol in the PwCOVID group, clinical parameters were recorded at rest, with people sitting on a chair: mean blood pressure was 130/75 (±19/±9) mmHg, heart rate 78 (±14) bpm, respiratory rate 19 (±4) and oxygen saturation 96% (±2). The same measurement was performed after the study protocol and no significant difference was found in these clinical parameters (at least, *p* > 0.52).

### 3.1. Cognitive Task

In the single counting task, PwCOVID performed similarly to HS, as shown by the CCR (0.49 versus 0.52, respectively, *p* = 0.61 with a negligible effect size). All subjects were able to perform this cognitive task.

### 3.2. Motor Task and Dual-Task: Stabilometric Assessment

Figure 1 shows the results of the static balance assessment in PwCOVID and HS. As expected, two-way ANOVA revealed an overall effect of the introduction of the second task on the mean sway path with EO (F(1,59) = 60.10, *p* < 0.0005). There was a difference between PwCOVID and HS (F(1,59) = 15.11, *p* < 0.0005), but the interaction did not reach significance (F(1,59) = 0.48, *p* = 0.49). The introduction of the second task also had an overall effect on the mean sway area with EO (F(1,59) = 21.31, *p* < 0.0005). There was a difference between groups (F(1,59) = 4.30, *p* < 0.01), but the interaction did not reach significance (F(1,59) = 0.41, *p* = 0.52).

In EC conditions, two-way ANOVA showed an overall effect of the introduction of the second task on mean sway path (F(1,59) = 17.41, *p* < 0.0005). There was a difference between groups (F(1,59) = 14.55, *p* < 0.0005), but the interaction did not reach significance (F(1,59) = 0.03, *p* = 0.87). The introduction of the second task did not reveal effects on the mean sway area with EC (F(1,59) = 3.67, *p* = 0.06).

DTC did not show significant differences between groups, except for the DTC of sway area with EO that was higher for HS (*p* < 0.05, *d* effect size = 0.58). However, the CCR was significantly better in HS compared to PwCOVID, both in EO (*p* < 0.005) and EC (*p* < 0.005) conditions, and with a large difference between groups (*d* effect size >0.80) for both visual conditions (Table 2).

Finally, there was a significant linear correlation between CCR EO and CCR EC, both in PwCOVID (r^2^ = 0.91, *p* < 0.0005) and in HS (r^2^ = 0.93, *p* < 0.0005) (Figure 2), and the slopes did not differ between groups (*t*-test on the regression slopes, t = 0.09, *p* = 0.89).

### 3.3. Motor Task and Dual-Task: Timed Up and Go Test

Regarding dynamic balance (Figure 3), two-way ANOVA did not show any effect of the introduction of the second task on the mean time employed to perform the TUG test (F(1,59) = 0.99, *p* = 0.32). However, there was a significant difference between the overall performance of PwCOVID and HS (F(1,59) = 14.16, *p* < 0.0005), but no significant interaction (F(1,59) = 0.29, *p* = 0.59).

The TUG test was able to detect the cognitive/motor interference in PwCOVID with respect to HS, as revealed by the significant difference in the CCR (0.36 versus 0.58, *p* < 0.005), with a large effect size (*d* effect size = 1.00). DTC was 16.09 in PwCOVID versus 9.43 in HS (*p* = 0.12), with a small-to-moderate effect size (*d* effect size = 0.40).

## 4. Discussion

To the best of our knowledge, this is the first study that assessed the effect of dual-tasks on stabilometry in PwCOVID. Not unexpectedly, stabilometry, as well as TUG test, performance were worse in PwCOVID compared to HS. Indeed, previous studies showed that PwCOVID, even those who had a severe but not critical form of COVID-19 in the acute phase, still had important COVID-19 sequelae after a month in rehabilitation [45,47].

However, the impact of adding a second cognitive task on balance was similar between groups. The main difference we found between PwCOVID and HS was the ability to count backwards during the dual-task condition, even if their ability to perform this cognitive task while seated was similar. Although PwCOVID had a normal score on the MMSE [33] and a similar ST cognitive performance to HS, they showed lower CCR values in the dual-task, indicating that PwCOVID had more difficulty to perform a cognitive task whilst performing another motor task. To our knowledge, no other studies have investigated the CCR in PwCOVID. Even if it is well-known that counting backwards during a balance test requires attentional resources that are particularly difficult for those with the poorest balance [48], in our study, the tendency of PwCOVID to favor their balance performance instead of properly performing the cognitive task was quite unexpected. Recent studies showed that the addition of a cognitive task to a gait or balance task amplifies the gait variability and results in greater postural sway in both a population of physically fit elderly people, as well as in subjects affected by neurologic diseases [49]. This behavior, usually called the posture-second strategy, was the one observed also in the HS enrolled in this study. In fact, during the DT trials, HS increased their body oscillations on the stabilometry and slowed down their execution of the TUG test in order to properly perform the cognitive task required. On the other hand, although PwCOVID recruited in this study had a normal MMSE score, the difficulty of the cognitive task could have affected their choice to give priority to the balance task. This possible explanation is corroborated by the findings of Maclean et al. [50]; these authors studied the interference of dual-tasks in healthy elderly people and found that the more complex the cognitive task added, the less patients prioritize it over the motor task, even if explicitly requested by the assessors. In our study, participants were not instructed about which task they had to prioritize, so PwCOVID might have unconsciously used a posture-first strategy to avoid losing their balance. Therefore, it is conceivable that PwCOVID left aside the correct execution of the cognitive task in favor of control over their balance, which was under threat from a heavier demand on their attentional resources.

Our results are aligned with those of Bergamin et al. [51] who reported an increase in the length and in the area of the sway during a spoken mental arithmetic task. This increase could be ascribed to different non-mutually exclusive causes. First, the respiratory muscle activity in relation to vocalization may increase the perturbation of postural sway [52]. Second, the increase in sway length alone may be related to the use of oculomotor activity as unintentional attempts to increase arousal by self-generated body movement. Finally, as demonstrated by Holmes et al. [53], a simple monolog (e.g., describing a familiar place) determined such a significant effect of cognitive load as to modify the postural stability. However, in our study, the respiratory rate assessed before and after each static and dynamic assessment (not shown in results) did not reveal any significant change in the breathing rate after the evaluation trials.

It could be suggested that another reason affecting the performance of PwCOVID in stabilometry is the muscle weakness induced by hospitalization and the reduction of mobility. Nevertheless, Morasso and Schieppati et al. [54] reported that the level of muscle activity during standing quietly is low. Moreover, in patients with respiratory disease, no relationship was found between balance performance and muscle strength of lower limbs [55]. In the same way, the abnormalities of gait observed during the TUG test do not appear to be mainly attributable to weakness or deconditioning. The test has a short duration and does not require a particular effort. In fact, while poor balance control has been reported to highly contribute to a longer duration of different phases of the TUG test, the strength of lower limbs appears to influence only the performance of the sit-to-walk phase of the TUG test [56]. Therefore, it is plausible that the poor cognitive performance achieved by the PwCOVID during the TUG test highlights that a simple verbal task can affect an automatic task such as gait and share complex neural pathways connecting different brain regions which are interlinked with those controlling gait [57].

There were several limitations in this study. First, the pragmatic design, with a convenience sample of PwCOVID and stratified random sample of HS generated from our pre-existing database, in addition to the absence of blinding of the assessors, may have introduced selection biases. Second, the nature of the study to enroll patients in a single COVID-19 ward of a rehabilitative institute may have not captured the entire spectrum of COVID-19 survivors. Moreover, these PwCOVID had a long hospitalization period and, even if we did not assess the presence of post-traumatic stress symptoms in these COVID-19 survivors, we can suppose that it is present and might have interfered with subjects’ performance. Another limitation was the use of MMSE instead of the MoCA, a scale that was suggested to be more sensitive in the detection of neurocognitive disorder, as it assesses executive function and visuospatial abilities [58,59]. This choice was made due to the longer time required by the MoCA; the complexity of the COVID-19 ward and the limited time available for the research activity prompted the researchers to use the MMSE.

Future studies might perform a long-term assessment, with the same evaluations, in order to investigate the evolution of the performance of PwCOVID. Moreover, we suggest also collecting data on subjects’ strength, in particular that of lower limbs, to verify the interference between strength and the performance of balance and gait. A prospective collection of falls data, made through a telephonic follow-up, might also be useful to understand the prediction power of the assessments performed. In this line, as to date no data currently exist about TUG test cut-off for risk of fall in PwCOVID, future studies might find the appropriate value.

## 5. Conclusions

In conclusion, our results suggest that severe illness due to COVID-19 leads to manifest cognitive deficit when subjects are required to perform simultaneous tasks, even in those who did not show clear cognitive impairments in common cognitive tests. Therefore, the findings of the present study encourage the use of a dual-task evaluation in PwCOVID, with specific attention to the assessment of both the motor cost and the cognitive cost response in order to detect the impact of dual-tasks on their performance. 

## Figures and Tables

**Figure 1 ijerph-19-10644-f001:**
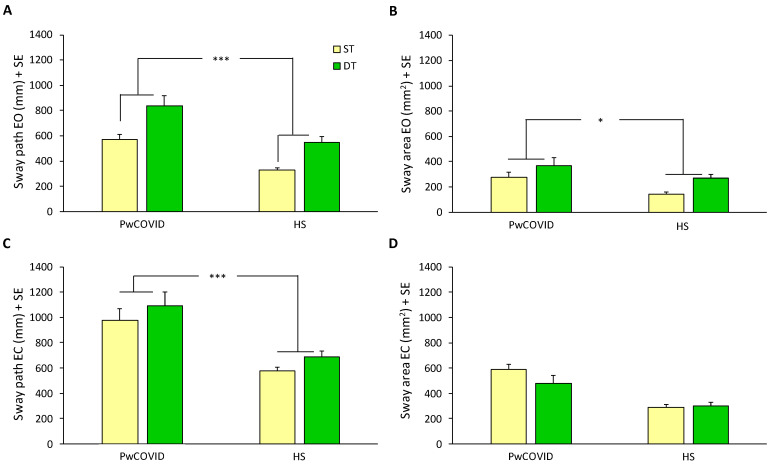
Sway path and sway area in EO and EC conditions, for PwCOVID and HS, with the ST vs. DT. PwCOVID and HS differed, as did the ST versus the DT, considering both groups together (not shown in the figure), except for the sway area with EC. Statistical analysis was performed with ANOVA, applying Bonferroni’s correction. *, *p* < 0.01; ***, *p* < 0.0005. Abbreviations: ST, single-task; DT, dual-task; PwCOVID, people with COVID-19; HS, healthy subjects; EO, eyes open; EC, eyes closed; SE, standard error.

**Figure 2 ijerph-19-10644-f002:**
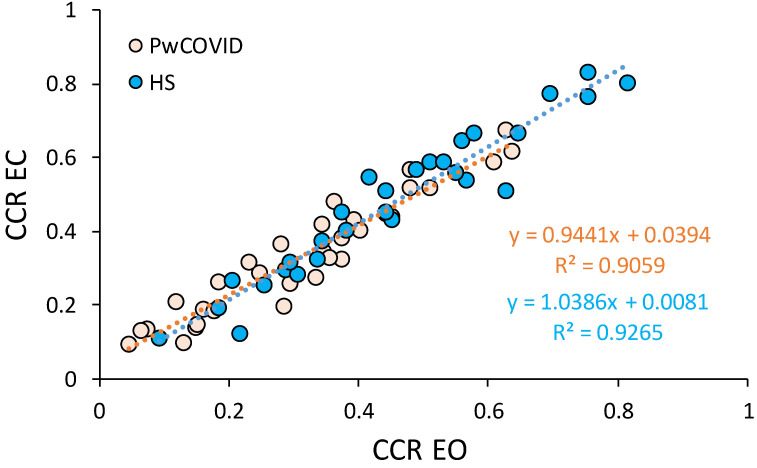
Relationship between CCR EO and CCR EC in PwCOVID and HS in stabilometric assessment. Subjects were instructed to count backwards in threes; number of total subtractions and errors were recorded to calculate each CCR, separately for each visual condition. Orange dots represent PwCOVID and blue dots HS. The relationship between CCR in the different visual conditions shows a linear correlation and the slopes of the fit line across each group are similar (orange and blue line for PwCOVID and HS, respectively). Abbreviations: PwCOVID, people with COVID-19; HS, healthy subjects; CCR, correct cognitive response; EO, eyes open; EC, eyes closed.

**Figure 3 ijerph-19-10644-f003:**
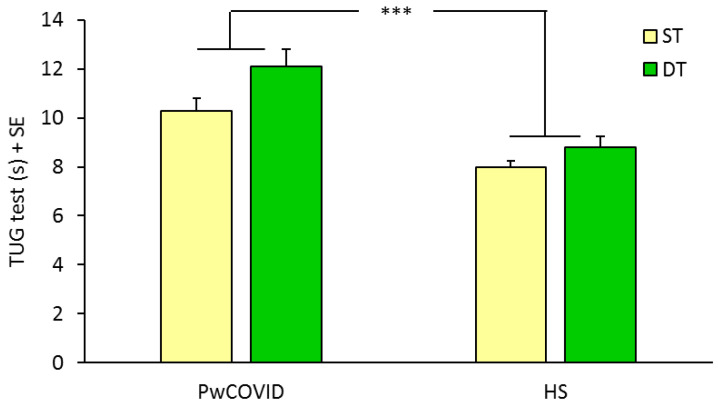
Mean time employed to perform the TUG test in ST and DT conditions. ANOVA showed a significant difference between the overall performance of PwCOVID and HS. ***, *p* < 0.0005. Abbreviations: PwCOVID, people with COVID-19; HS, healthy subjects; ST, single-task; DT, dual-task; TUG, Timed Up and Go; SE, standard error.

**Table 1 ijerph-19-10644-t001:** Characteristics of patients and healthy control: comparison between groups.

		PwCOVID		HS		*p*
		(n = 31)		(n = 30)		
		Mean	SD	Mean	SD	
N. males; females		24; 7		17; 13		0.08
(% female)		23		43		
Age (years)		68.65	10.15	68.90	6.47	0.91
Body Weight (kg)		74.35	14.01	70.03	12.95	0.17
Height (cm)		1.70	0.07	1.66	0.08	0.05
Body Mass Index		25.70	3.84	25.72	3.97	0.96
NIS total score		1.29	1.49	1.27	1.55	0.86
MMSE score		27.30	1.84	28.13	1.34	0.09
CIRS total score		23.24	3.61			

Abbreviations: NIS, Neuropathy Impairment Score; MMSE, Mini Mental State Examination; CIRS, Cumulative Illness Rating Scale; SD, standard deviation; PwCOVID, people with COVID-19; HS, healthy subjects.

**Table 2 ijerph-19-10644-t002:** Comparison between PwCOVID and HS in dual-task cost and correct cognitive response.

	PwCOVID (n = 31)		HS (n = 30)		*p*	*d*
	Mean	SE	Mean	SE		
DTC EO sway path	47.00	6.75	65.46	9.30	0.11	0.41
DTC EO sway area	55.45	15.97	111.69	19.34	<0.05	0.58
DTC EC sway path	17.88	4.61	19.86	4.69	0.76	0.08
DTC EC sway area	−4.97	7.13	8.95	7.76	0.19	0.34
CCR EO	0.31	0.03	0.45	0.03	<0.005	0.80
CCR EC	0.33	0.03	0.48	0.04	<0.005	0.83

Abbreviations: PwCOVID, people with COVID-19; HS, healthy subjects; DTC, dual-task cost; CCR, correct cognitive response; EO, eyes open; EC, eyes closed; SE, standard error; *d*, Cohen’s *d* effect size.

## Data Availability

The data that support the findings of this study are available from the corresponding author, I.A., upon reasonable request.

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
