# Peer review of "The Coronavirus Footprint on Dual-Task Performance in Post-Acute Patients after Severe COVID-19: A Future Challenge for Rehabilitation"

_ijerph, 2022, doi:10.3390/ijerph191710644_

Round 1
Reviewer 1 Report
Dear authors
The clinical message is important, but the paper need to be improved.
Main point: The introduction and discussion need to be revised. Many paragraphs seem "thrown" in the text. The sequence in which they appear shows no connection between them. This does not allow the creation of a line of reasoning that leads the reader to understand the importance of the topic and the development of the research.
My suggestions and recommendations are explained in further detail below.
Introduction
1) I felt like it needed more information on COVID-19 and the study basis in general (executive function, balance, risk of falls, aging and rehabilitation).
2) Line 34-38 / 39-44: This phenomenon needs to be better explained. All this needs to be explained, with a narrative that makes the reader go through a whole line of reasoning until reaching the objective of the study and, then, understanding its importance.
3) Line 68-74: this sentence is confusing. Please rewrite.
Methods
1) Line 91: one of its inclusion criteria was age over 18 years, but the mean age of both groups was 68 years. This seems meaningless, since the theme is related to aging.
2) Line 103-104: " For all participants, general characteristics such as age, sex, weight, height and body mass index (BMI) were collected.”
I am assuming the BMI was calculated from these anthropometric questions. Is this correct? If so, I recommend that you inform that the calculation was based on self-report and it wasn't measured. If not, you should describe you have measured the weight and height and have calculated the BMI.
3) This scale (Cumulative Illness Rating Scale) appears in Table 1, but is not mentioned in the methods.
4) Item Test procedure: Was there any blinding of the assessors? The HS group was assessed by the same assessors as well?
5) there was no randomization process. The samples were convenience samples and evaluated at very different times (database). This needs to be made clear in the text
6) were the patients previously familiarized with the tests (in addition to the TUG) to minimize the learning effect?
7) Line 166-170 - I was confused by the passage: “Morelli et al., in their assessment of DT performance during the TUG test in PwCOVID, found a difference of 3.8 ± 5.2 s between individuals with COVID vs. with chronic lung disease [22]. This corresponds to a Cohen’s d effect size of about 0.7. Setting the significance at 5% and statistical power at 80%, we therefore required a sample size of 27 subjects for each group [35]. I think it would be nice if this could be explained better, since the comparison group is individuals with chronic lung disease (and not with healthy controls).
1) Line 183-185 - This sentence: “In order to investigate the clinical meaning of differences between PwCOVID and HS, Cohen's d effect size was calculated, with a small effect defined as 0.01, a medium effect as 0.06, and a large effect as 0.14 [35].” I didn't find this information in the reference you mentioned (and these values are not compatible with those presented in the tables). Please, check the inconsistency and address it accordingly.
Results / Discussion:
For the results / discussion I add some points for reflection:
1) Emphasizing intragroup differences (increase in performance with the addition of DT seems obvious to me) and mentioning a tendency to be higher (p=0.12 – line 260) seems to overestimate the data. I suggest that the authors review the form of presentation.
2) There was no interaction effect – this is the most important thing to consider in both chapters.
3) Line 293-296: “This value, although still not sufficient to demonstrate a clear increase in risk of falls (as the authors identified a DTC >20% for gait velocity to indicate a destabilizing effect of the DT [40]), is nevertheless a high value for people under-70 years old, almost double that of the HS”.
The value is not sufficient to demonstrate an increased risk of falls. I suggest removing this information.
4) Could the period of hospitalization have interfered with these results? the Stress experienced by patients? immobility syndrome? consider addressing these topics.
5) You should discuss about the limitations of your study (type II error, selection bias, for instance).
6) Thinking about the clinical part, what is the message (clinical recommendations/implications to the rehabilitation)?
7) Line 335-337: “From our results, it appears that these people, even though they score in the normal range on cognitive clinical tests, may have cognitive impairments that could lead to risk of falls in their return to daily life at home”. The conclusion is too long and also needs to be revised. “It appears” is to overestimate the final idea intended for the reader. I would be more caution in your conclusions.
Reviewer 2 Report
What was the average length of stay in the unit for PwCOVID? Sounds like they were still in the hospital day 25. Was this one of the inclusion criteria - minimum length of stay?
What was the reasoning for when in their inpatient stay this test was performed? Why day 25?
It is not clear how healthy subjects were recruited for this study at a COVID-19 Rehab Unit?
What was the purpose of collecting this data on HS in previous years in your unit?
How were healthy subjects recruited?
Test Procedure:
Was a practice trial provided?
Minor comments:
Please add a space between p < 0.005 and between '=' sign and number throughout the text
Was there a time interval between the two tests
Sounds like there were some patients that were not on supplemental oxygen. This can impact their balance performance . also how was the supplemental o2 tank carried during TUG? Was it that the patient carried it themselves?
How was SpO2 monitored continuously using a pulse ox? Were patients asked to stop during TUG ? How was SpO2 monitored during the test?
Again, not sure how RR was measured during the test.
Need more specifications on when and how many times these parameters
(RPE, SpO2, RR) were checked for patients.
